# *NUDT15* Pharmacogenetics in Acute Lymphoblastic Leukemia: Synthesizing Progress for Personalized Thiopurine Therapy

**DOI:** 10.3390/medsci13030112

**Published:** 2025-08-05

**Authors:** Isfahan Shah Lubis, Kusnandar Anggadiredja, Aluicia Anita Artarini, Nur Melani Sari, Nur Suryawan, Zulfan Zazuli

**Affiliations:** 1Department of Pharmacology-Clinical Pharmacy, School of Pharmacy, Institut Teknologi Bandung, Bandung 40132, Indonesia; lubisisfahan@gmail.com (I.S.L.); kusnandar_a@itb.ac.id (K.A.); 2Department of Pharmaceutics, School of Pharmacy, Institut Teknologi Bandung, Bandung 40132, Indonesia; anita.artarini@itb.ac.id; 3Division of Hematology-Oncology, Department of Child Health, Faculty of Medicine, Universitas Padjadjaran/Dr. Hasan Sadikin General Hospital, Bandung 40161, Indonesia; nur.melani.sari@unpad.ac.id (N.M.S.); nursuryawan@gmail.com (N.S.)

**Keywords:** NUDT15, pharmacogenetics, thiopurines, acute lymphoblastic leukemia (ALL), myelosuppression, personalized medicine, 6-mercaptopurine

## Abstract

The management of acute lymphoblastic leukemia (ALL), the most common pediatric malignancy, critically relies on thiopurine therapy, such as 6-mercaptopurine (6-MP), during the maintenance phase. However, significant inter-individual response variety and high risk of myelosuppression often disrupt therapy efficacy. Pharmacogenetics offer crucial strategies to personalized therapy. While thiopurine methyltransferase (*TPMT*) was initially the primary focus, the discovery of *nudix hydrolase 15* (*NUDT15*) appears as a more comprehensive determinant of thiopurine intolerance. This review aims to consolidate and critically evaluate the advancement achieved in unraveling the biological mechanism and clinical significance of *NUDT15* pharmacogenetics in thiopurine therapy. Foundational studies showed the vital role of *NUDT15* in the detoxification of active thiopurines, with common genetic variants (for instance, p. Arg139Cys) significantly disrupting its activity, leading to the accumulation of toxic metabolites. Observational studies consistently associated *NUDT15* variants with severe myelosuppression, notably in Asian populations. Recent randomized controlled trials (RCTs) confirmed that *NUDT15* genotype-guided dosing effectively reduces thiopurine-induced toxicity without interfering with the therapeutic outcome. Despite these advancements, challenges remain present, including the incomplete characterization of rare variants, limited data in the diverse Asian populations, and the need for standardized integration with metabolite monitoring. In conclusion, *NUDT15* pharmacogenetics is essential for improving patient safety and thiopurine dosage optimization in the treatment of ALL. For thiopurine tailored medicine to be widely and fairly implemented, future research should focus on increasing genetic data across different populations, improving the dose adjustment algorithm, and harmonizing therapeutic guidelines.

## 1. Introduction

Acute lymphoblastic leukemia (ALL) is the most common pediatric malignancy, requiring aggressive and immediate therapeutic intervention [1]. Central to its successful management, chiefly during the crucial maintenance phase, 6-mercaptopurine (6-MP) administration becomes critical [2]. Despite being highly effective in eradicating residual leukemic cells and preventing relapse, thiopurine therapy is known to have wide inter- and intra-individual variability response and significant risk of severe adverse drug reactions (ADRs), notably myelosuppression [3,4,5]. This variability leads to considerable challenges, often requiring dose reductions or therapy interruption, which disrupts therapy efficacy and patient clinical outcomes.

The advent of pharmacogenetics has revolutionized our approach to mitigating such drug-related toxicity by identifying genetic predispositions. In the context of thiopurine therapy, initial research focused on the polymorphism of thiopurine methyltransferase (*TPMT*) [6,7,8,9,10,11,12,13,14,15,16,17,18,19,20,21,22]. While TPMT variants are a major genetic determinant for toxicity in European populations (e.g., associated with 31% of severe hematotoxicity cases [23]), they are very rare in Asian cohorts. For instance, a study in Chinese children with ALL found a TPMT minor allele frequency of only 2.9%, compared to 15.7% for NUDT15 variants [24]. Subsequently, extensive research also explored the role of other key enzymes, such as cytosolic 5′-nucleotidase II (NT5C2), which was found to be a significant contributor to thiopurine metabolism and drug resistance, particularly in relapsed ALL [25,26,27,28,29,30,31]. However, the discovery of nudix hydrolase (NUDT15) as an important enzyme in the metabolism of thiopurine marks a significant breakthrough because its variants are highly prevalent in Asian populations and, as demonstrated in a Korean study, showed a predictive sensitivity of 89.4% for early leukopenia, compared to only 12.1% for TPMT variants [32], offering a more comprehensive explanation for interindividual variability, especially in populations where *TPMT* variants do not fully explain thiopurine intolerance [24,33,34,35,36,37]. Since then, variations in the *NUDT15* gene have been unambiguously associated with the shift of enzymatic activity and the increased risk of myelosuppression, profoundly influencing the personalized strategy of thiopurine dosing.

Against this backdrop, this comprehensive review aims to consolidate and critically evaluate the progress made to date in unravelling the biological mechanism and clinical significance of *NUDT15* pharmacogenetics. Beginning with theoretical basis of acute lymphoblastic leukemia and thiopurine metabolism, this review discusses the molecular and functional basis of *NUDT15*, including the explanation of key genetic variants and their impact. Subsequently, this review synthesizes clinical evidence from observational studies and randomized controlled trials (RCTs), offering insight into the real-world implication of *NUDT15* genotyping for patient safety and therapy efficacy. Ultimately, this review addresses current challenges and future perspectives in the field of *NUDT15* pharmacogenetics.

## 2. Fundamentals of Thiopurine Pharmacogenetics in Acute Lymphoblastic Leukemia

### 2.1. Acute Lymphoblastic Leukemia

Acute lymphoblastic leukemia (ALL) is defined as a hematological malignancy marked by the rapid proliferation of immature lymphoid cells, known as lymphoblasts, in bone marrow and peripheral blood [38]. Its aggressive progression and acute onset classify it as an acute condition, thus requiring immediate medical intervention. The pathophysiology of ALL involves clonal expansion of lymphoid progenitor cells, which results in the significant accumulation of lymphoblasts in bone marrow. This perturbs normal hematopoiesis, leading to cytopenia, anemia, thrombocytopenia, and leukopenia [39]. Based on the affected lymphoid cell lines, ALL is subclassified into B-cell ALL (B-ALL) and T-cell ALL (T-ALL).

Despite occurring in all age groups, ALL predominantly affects children, with approximately 80% of ALL survivors being pediatric patients. Among pediatric malignancies, ALL is the most common cancer, with the highest incidence observed in children aged 1–5 years old [40]. Additionally, the incidence of ALL is 30% higher in boys than girls [41]. The risk factors of ALL are multifactorial and involve a complex interplay between environmental factors such as high levels of pollutants [42], genetic factors such as polymorphisms of relevant enzyme-coding genes [43], and chance. A series of clinical manifestations of ALL arise from the infiltration of lymphoblasts into the bone marrow and peripheral blood. Consequently, the primary detectable manifestations of ALL are hematological, encompassing anemia, which eventually leads to fatigue, paleness, and weakness. Along with anemia, thrombocytopenia and leukopenia are also common clinical findings in ALL patients. Thrombocytopenia can lead to bruising, petechiae, and an increased bleeding tendency, while leukopenia results in a higher susceptibility to infection [44,45].

The pharmacological therapy of ALL consists of three phases: induction (remission), consolidation, and maintenance, with each phase aiming for different outcomes (Figure 1). Daily 6-mercaptopurine (6-MP), along with weekly methotrexate, is a standard combination in the maintenance phase, which aims to eradicate remaining leukemic cells and to prevent disease recurrence. The vital role of 6-MP in the therapeutic efficacy of maintenance phase is demonstrated by the increased risk of relapse in cohorts with incomplete or truncated maintenance phase therapy [46,47]. Due to various significant adverse drug reactions and the aggressive nature of ALL therapy 6-MP administration should be accompanied by meticulous drug dosing in order to ensure event-free survival.

### 2.2. Metabolism and Mechanism of Action of Thiopurines

Thiopurines are prodrugs that undergo a series of enzymatic reactions, converting them into active metabolites responsible for their therapeutic effects (Figure 2). Azathioprine (AZA) is a parent drug from which 6-MP is derived via non-enzymatic hydrolysis. Following conversion, 6-MP is metabolized through three primary pathways involving key enzymes. TPMT methylates 6-MP to form methylmercaptopurine (MMP, also known as MeMP). Xanthine oxidase transforms 6-MP into thiouric acid, which is subsequently eliminated. Lastly, hypoxanthine–guanine phosphoribosyltransferase (HGPRT) converts 6-MP into thioinosine monophosphate. This intermediate is then further transformed into thioguanine monophosphate (TGMP) by inosine monophosphate dehydrogenase (IMPDH) and guanosine monophosphate synthetase (GMPS). The metabolism of thioguanine (TG) is comparable to that of 6-MP. TGMP can then be converted into the active 6-TGTP by kinases. 6-TdGTP may also be produced through various enzymatic mechanisms. Lastly, the thiopurine 6-TdGTP and 6-TGTP active metabolites are hydrolyzed by NUDT15 [48]. Another enzyme, inosine triphosphate pyrophosphatase (ITPA), is also involved in this pathway, converting thio-ITP to thio-IMP, thereby influencing the pool of thiopurine metabolites.

The cytotoxic effect of thiopurine drugs, including 6-MP and AZA, mainly arises from their incorporation into DNA which affects rapidly proliferating cells. When incorporated into DNA, the nucleotides derived from thiopurine metabolism interfere with the normal cellular purine metabolism, leading to base mispairing during DNA replication. This incorporation produces aberrant base pairs, which are subsequently recognized and processed by the cellular mismatch repair (MMR) system. The recognition of these adducts can trigger a series of DNA damaging reactions, leading to apoptosis and cell cycle arrest, which are key therapeutic targets in ALL [51]. Furthermore, thiopurines exert their cytotoxic effects via other non-canonical mechanisms beyond incorporation into DNA. For instance, thiopurines’ antiproliferative action stems from their capacity to cause oxidative stress, mitochondrial malfunction, and ATP depletion, leading to irreversible energetic collapse and necrotic cell death [52]. Additionally, thioinosine nucleotides and their methyl derivatives (MeMPs) can inhibit important enzymatic functions such as *de novo* purine synthesis, which is necessary for lymphoid cell proliferation [50,53].

### 2.3. Nucleoside Diphosphatase-Linked Moiety X-Type Motif 15 (NUDT15)

#### 2.3.1. Gene Structure and Enzymatic Function of NUDT15

Comprising three exons, the *NUDT15* gene codes for the enzyme NUDT15 which is a member of the NUDIX hydrolase family. NUDT15 hydrolyzes nucleoside diphosphates to nucleoside monophosphate via the highly conserved NUDIX box. The NUDT15 protein forms a homodimer, unlike many other proteins in the NUDIX family [54]. NUDT15 plays a pivotal role in the metabolism of thiopurine drugs primarily by hindering their toxic effects without impeding their pharmacological activity. NUDT15 metabolizes thiopurine drugs by hydrolyzing thiopurine nucleotides, specifically converting 6-thioguanine triphosphate (6-TGTP) and 6-thiodeoxyguanine triphosphate (6-TdGTP) into their inactive forms, 6-thioguanine monophosphate (6-TGMP) and 6-thiodeoxyguanine monophosphate (6-TdGMP), respectively [55]. These monophosphate forms are structurally incapable of being incorporated into DNA and RNA. This enzymatic activity is indispensable for maintaining the safe intracellular levels of bioactive thiopurine metabolites. To protect normal hematopoietic function while still enabling the drug’s therapeutic efficacy in targeting malignancies, NUDT15 helps to mitigate the cytotoxic effects associated with thiopurines by blocking the incorporation of these toxic metabolites into DNA. For this reason, a decline in NUDT15 enzymatic activity may lead to an accumulation of toxic nucleotides, which in turn increases the risk of adverse drug reactions, such as myelosuppression and organ toxicity [56]. Considering these, the administration of 6-MP should be accompanied by meticulous drug dosing to ensure event-free survival. To establish a more informed and individualized pharmacotherapy, pharmacogenetic factors, especially relating to relevant variants, including nudix hydrolase 15 (NUDT15), should be considered [57,58].

In addition to genetic polymorphism, the regulation of *NUDT15* expression in transcriptional level has been linked to the resistance to thiopurines. Several studies show that the downregulation of mRNA expression and NUDT15 protein, despite genetic variations, could cause an increase in the accumulation of active thiopurine metabolites and drug resistance in leukemic cells. Furthermore, the expression of *NUDT15* is also subject to post-transcriptional regulation, with evidence showing the potential role of microRNA (miRNA) in modulating the NUDT15 protein level [59]. These findings emphasize that the impact of NUDT15 in thiopurine metabolism is a multilayer process that involves not only inherent genetic variants but also dynamic mechanisms of regulation.

#### 2.3.2. Functional Impact of Key Genetic Variants

Several studies have shown that variability in NUDT15 enzymatic activity, and therefore the risk of toxicity caused by thiopurine, is largely proceeding from the inherited genetic variations within the *NUDT15* gene. A schematic representation of the *NUDT15* gene structure and the location of these key variants is provided (Figure 3). One of the most pivotal research efforts was conducted by Moriyama et al. [60], which successfully elucidated the relationship between germline *NUDT15* variation and NUDT15 enzymatic activity, particularly concerning thiopurine toxicity across populations of diverse ancestries. They have identified four *NUDT15* coding variants in Guatemalan, Singaporean, and Japanese populations (p.Arg139Cys, p.Arg139His, p.Val18Ile and p.Val18_Val19insGlyVal), constituting five haplotypes. All variants significantly altered thiopurine metabolism due to a considerable decline in NUDT15 enzymatic activity, resulting in a 74.4–100% loss of nucleotide diphosphatase activity and consequently leading to thiopurine intolerance across all the populations [60]. They also identified three novel coding variants, p.Arg34Thr, p.Lys35Glu, and p.Gly17_Val18del in five children, all of which led to a complete loss of NUDT15 enzymatic activity, highlighting a potentially significant area for future research [61]. Nevertheless, due to a limited number of studies and evidence regarding these new variants, this review will focus on the five previously established coding variants.

## 3. *NUDT15* Pharmacogenetics: Unraveling the Molecular and Functional Basis

Research on the role of NUDT15 in the metabolism of thiopurine has offered us substantial insights, starting from fundamental exploration of its mechanism to translational investigations. The work of Valerie et al. [55] became one of the initial significant contributions, not only explaining the role of NUDT15 in hydrolyzing the active metabolites of thiopurine but also offering a critical perspective on the mechanism behind thiopurine sensitivity in patients bearing the R139C variant. The research team argued that protein instability, which causes protein degradation, was the main driver of thiopurine sensitivity, instead of a mere decrease in enzymatic activity. Hence, this breakthrough finding has introduced another layer of complexity, often overlooked in simpler interpretations. This multifaceted study, which employed a firm combination of structural, enzymatic, and cellular analysis, presented a solid cornerstone to comprehend the complex interplay between *NUDT15* genotype and response to thiopurines.

Building upon this cellular foundation, in vitro studies have directly linked the *NUDT15* variation and the shift in drug metabolism confirmed in leukemic cell lines. The significant rise in the level of intracellular DNA-TG and chemosensitivity in this model demonstrated a direct proof regarding the functional consequences of *NUDT15* genetic variation in the context of relevant cancer. Nevertheless, the inherent simplification of the in vitro system requires attention when this finding is translated into a clinical context, which is undoubtedly more complex. Furthermore, the identification of *NT5C2* and *PRPS1* gene variation as additional factors demonstrate that the pharmacogenetics of thiopurine metabolism has expanded beyond a single gene, highlighting the urgency of a more integrated model [62].

The transition to in vivo studies, especially those employing knock-in mice models mimicking clinically relevant variants, marks a major step to more clinically relevant findings. The use of the *Nudt15* knock-in mice model, which mimics the clinically established *NUDT15* variant of p.Arg139Cys, has revealed a significant impact of NUDT15 deficiency on hematopoietic stem cells, manifested by the increase in thiopurine-mediated DNA damage and exhaustion via Trp53 networks. This study provides a strong mechanistic explanation for clinically observed myelosuppression [63]. Similarly, another study employing *Nudt15* knock-in mice reinforces the association between hematotoxicity and p.Arg139Cys variation, suggesting a direction to explore genotype-guided pharmacotherapy strategy [64].

To narrow the gap between animal model studies and the clinical relevance of their findings, Nishii et al. [65] conducted research to directly evaluate the genotype-guided dosing in preclinical models, yielding important translational data. Their findings, that the *Nudt15* knock-out mice model showcased an increase in toxicity, which could be mitigated by dose adjustment without sacrificing the antileukemic efficacy of thiopurines on a xenograft model, strongly supports the clinical rationale of personalized thiopurine therapy. However, a critical consideration about this research lies in the potential nonexclusive function of NUDT15 in the biosynthesis of nucleotides, as shown by the presence of 22 *NUDT* genes with various overlapping hydrolase activities. Therefore, the observed toxicity in complete knock-out *Nudt15* models may reflect a disruption beyond thiopurine metabolism alone, which potentially affects the fundamental cellular process compensated for by other NUDT family members in individuals with partial NUDT15 deficiency. Understanding the complete spectrum of the physiological role of NUDT15, through comprehensive metabolic profiling, is crucial to accurately translate these preclinical findings into the diverse genetic landscape of patient populations.

Contributing to efforts in explaining the functional consequences of *NUDT15* variation, a study by Man et al. [66] employed structural biophysics techniques to investigate p.Arg139Cys and p.Val18Ile mutations regarding thiopurine intolerance. Their key findings on the temperature-dependent destabilization of the catalytic site of the NUDT15 enzyme offer a molecular-level explanation for the decreased enzymatic activity of NUDT15 concluded from previous in vivo studies. While this in vitro work grants us a valuable mechanistic insight on how specific mutations might impair enzyme stability, a factor that potentially affects the efficacy of thiopurine detoxification systematically observed in animal models, its direct correlation with enzymatic activity in vivo and broader clinical spectrum of *NUDT15* clinical variants remain to be fully established. Further studies are required to clarify the differential impact on cancer versus normal cells to close the gap between these detailed molecular findings and direct clinical application.

The emergence of high throughput screening technology, exemplified by the work of Suiter et al. [67] in massively parallel characterization of variants, represents a significant advancement in our ability to comprehensively assess the functional consequences of major *NUDT15* variants. The strong correlation with clinical toxicity data lends considerable weight to this approach in order to predict individual risk of thiopurine toxicity. Nonetheless, the limitation in cytotoxicity-based screening methods, particularly for variants with moderate effects, indicates that our understanding of the complete spectrum of clinically relevant *NUDT15* variants is still evolving.

Building upon these endeavors, preclinical research on *NUDT15* pharmacogenetics, encompassing in vitro mechanistic studies and in vivo animal modeling on the effect of *NUDT15* variants on the metabolism and toxicity of thiopurines, has laid a critical foundation for understanding the functional consequences of *NUDT15* polymorphism. These efforts have moved this field beyond mere genetic associations to the explanation of underlying molecular mechanisms and the evaluation of potential genotyping-based drug dosing in controlled laboratory settings. However, the inherent complexity in translating these findings to human patients, including the influence of compensatory pathways and the needs of comprehensive functional characterization in a broader spectrum of variants, requires further investigation in clinical settings. Therefore, the subsequent section of this review will focus on observational studies and randomized controlled trials aiming to validate the clinical utility of *NUDT15* pharmacogenetics in predicting patient outcomes and guiding personalized thiopurine therapy in real-life scenarios.

## 4. The Clinical Impact of NUDT15 Variants: Insights from Observational Studies and RCTs 

Building upon the evolving understanding of *NUDT15* pharmacogenetics gained from in vitro and in vivo studies, the investigation of clinical relevance of *NUDT15* variants requires direct observation in human subjects. Observational studies and randomized controlled trials play a vital role in translating these fundamental insights into practical benefits for patients receiving thiopurine therapy. By investigating the relationship between *NUDT15* polymorphisms and clinical outcomes, specifically the incidence and severity of adverse drug reactions such as myelosuppression, these studies provide important evidence for the potential of genotype-guided drug dosing to enhance the safety and efficacy of therapy in real-world settings. While the body of evidence is still evolving, the existing research offers compelling insights, especially in particular ethnic groups where certain *NUDT15* variants are more prevalent and exert a pronounced effect on thiopurine metabolism.

The pronounced impact of *NUDT15* variants in the Asian population is a recurring theme. A prospective study by Zhou et al. [24] in Chinese children with ALL, which found *NUDT15*′s minor allele frequency (MAF) to be 15.7% compared to just 2.9% for *TPMT*, demonstrated a clear association between *NUDT15* variants and dose intensity. Patients with the homozygous *NUDT15* TT genotype tolerated an average dose intensity of only 60.27%, significantly lower than patients with the CC wild-type (94.24%). The *NUDT15* variant was also identified as an optimal predictor for leukopenia (OR: 3.62) and early-onset leukopenia (OR: 9.63) [24]. These findings are consistent across various Asian populations; a multi-ethnic Southeast Asian study showed *NUDT15* variants strongly predicted leukopenia in Chinese and Indian patients (with a combined OR of 33.80), but not in Malays, highlighting the need for population-specific analysis [35]. Furthermore, a large Canadian cohort of pediatric patients with IBD found that while TPMT variation was more prevalent in individuals of European ancestry (8.7%), NUDT15 variation was more prevalent in East Asian ancestry (16%), reinforcing the ethnic disparity in pharmacogenetic determinants [68].

The strong association between *NUDT15* variants and severe myelotoxicity is further solidified by meta-analysis data. A recent meta-analysis of 30 studies in Asian populations found that NUDT15 genetic polymorphisms were associated with a pooled odds ratio of 11.43 (95% CI 7.11–18.35) for early leukopenia and 16.35 (95% CI 10.20–26.22) for early neutropenia [69]. The analysis also identified specific alleles, such as *NUDT15 *3* and *NUDT15 *2*, as key genetic markers with significantly increased risks, demonstrating the potent and allele-specific impact of these variants on drug toxicity.

A prospective study by Wang et al. [70] in Chinese patients with inflammatory bowel diseases (IBD), which showcased a significant association between *NUDT15* R139C polymorphism and azathioprine-induced leukopenia, underlined the clinical relevance of this genotype in ethnic groups where this variant is more frequent. This finding is consistent with a retrospective case-control study by Kishibe et al. [71] in Japanese patients with dermatological conditions, which convincingly demonstrated that severe myelotoxicity was almost exclusively confined to individuals with homozygous R139C variants in their cohort. The consistency of these findings across independent studies in the East Asian population reinforce the argument for routine preemptive *NUDT15* genotyping in this demography to mitigate the risk of severe toxicity. Furthermore, the numerous case reports from various Asian countries [72,73,74,75,76,77] evidently depict the profound sensitivity to thiopurine in individuals with specific *NUDT15* genotypes, often resulting in severe and rapid myelosuppression, even at standard or low initial thiopurine dose. These anecdotal reports, while not providing statistical power, offer illustrative real-world examples that support the findings of larger observational studies.

Conversely, the lack of a statistically significant association in a prospective study by Correa-Jimenez et al. [78] involving Colombian children with ALL highlighted the importance of considering population-specific genetic architecture. The authors acknowledged the limited sample size as a potential factor, possibly hindering the detection of a true association, especially if the frequency of the risk allele is lower in their Latin American cohort compared to the East Asian populations, where a strong signal has been consistently detected. This discrepancy underlines the need for further research in various ethnic groups to establish the global clinical utility of *NUDT15* genotype testing.

A retrospective study by Kang et al. [79] in Korean pediatric patients with IBD offers a more detailed understanding by correlating the NUDT15 metabolizer status (normal, intermediate, and poor metabolizers) to time-to-leucopenia and suggesting lower 6-TGN targets to intermediate metabolizers. This research provides valuable insights, demonstrating that solely relying on 6-TGN monitoring, a common practice, may not be sufficient to protect patients with certain *NUDT15* genotypes from early toxicity. Nonetheless, the retrospective study design, along with the single-center design, limits the generalizability of their specific 6-TGN target recommendation to wider populations or other clinical settings. The exclusion of patients who discontinued AZA due to non-leukopenic adverse reactions may also introduce selection bias, potentially underestimating the broader impact of *NUDT15* variants on overall thiopurines intolerance.

The prospective study by Ju et al. [33] in Korean pediatric ALL patients aimed to investigate the clinical significance of DNA-TGN monitoring in the context of *NUDT15* variants, thereby contributing to the understanding of thiopurine management in this patient group. The key finding includes a positive correlation between DNA-TGN level and 6-MP dose, demonstrating that comparable metabolite levels could be achieved in *NUDT15* variant carriers with a lower dose of 6-MP. Nonetheless, the high variability observed in the DNA-TGN level among variant carriers highlights the necessity of meticulous and continuous monitoring, implying that a genotype-guided single dose adjustment may not be sufficient for precise management. Significant limitations in this study encompass low incidence of cytopenia, probably stemming from cautious dose increment, hindering the determination of a definitive and specific toxic level threshold of DNA-TG. Moreover, a short follow-up period prevents long-term evaluation of clinical outcomes such as relapse and survival. The inclusion of newly diagnosed and relapsed ALL patients without a detailed subgroup analysis also presents a potential confounder.

The clinical utility of *NUDT15* genotyping is further supported by a series of RCTs. An RCT study by Chang et al. [80] in Korean patients with IBD demonstrated a significant reduction in thiopurine-induced myelosuppression and leukopenia in several genotype groups where patients carrying *NUDT15* variants received a lower initial dose or alternative therapy. The robust study design provides strong evidence for the benefit of preemptive genotyping. Similarly, a study by Chao et al. [81] in Chinese patients with Crohn’s disease showed that genotype-guided dose optimization strategy in the *NUDT15* variant of p.Arg139Cys effectively lowered the incidence of thiopurine-induced leukopenia without sacrificing its efficacy. These interventional studies directly support the hypothesis generated by observational studies that *NUDT15* genotype-guided initial thiopurine dose adjustment may potentially enhance patients’ safety without diminishing the effects of thiopurines. A recent multicenter RCT by Zhou et al. [82] in Chinese pediatric patients with ALL further reinforces this evidence in the setting of hematology malignancy in pediatric patients, proving the clear benefit of 6-MP dose adjustment to suppress the incidence of myelosuppression using a far lower initial dose. However, the fact that this is one of the few RCTs in pediatric ALL underscores the need for more such studies to establish robust, evidence-based guidelines in this specific patient population.

Based on this robust body of evidence, clinical guidelines have been developed to integrate NUDT15 pharmacogenetics into routine practice. For example, the latest updates from the Clinical Pharmacogenetics Implementation Consortium (CPIC) provide specific genotype-based dosing recommendations. These guidelines suggest a significant dose reduction (e.g., up to 50%) for patients with NUDT15 heterozygous variants, while those with homozygous variants may tolerate only a small fraction of the standard dose (e.g., as low as 8%). The CPIC recommendations also specify that patients who are intermediate metabolizers for both TPMT and NUDT15 are recommended to begin with a starting dose of 20–50% of the normal dosage to mitigate the combined risk of severe toxicity [83]. The clinical relevance of this compound metabolizer phenotype is highlighted in a study on Brazilian patients with ALL, which found that 0.9% of the cohort were heterozygous carriers for both NUDT15 and TPMT non-function alleles. For these compound intermediate metabolizers, a large reduction in the initial thiopurine dosing is recommended due to a potential highly increased risk of toxicity [84].

While *NUDT15* and *TPMT* are the predominant genetic factors, a study in children with ALL from the Middle East found that ITPA variants also contribute to thiopurine intolerance. In this cohort, patients with two ITPA risk alleles had a median 6-MP dose intensity of 65.33%, a notable decrease from the 100% median for the wild-type. Although this effect was less significant than that of NUDT15 variants, which resulted in a median dose intensity of 33.33%, the findings highlight the importance of considering other genetic markers, such as ITPA, particularly in populations where their prevalence is higher [85].

The importance of population-specific pharmacogenetics is further highlighted by the research on underrepresented minor ethnic groups. The research on Chinese nationalities, including Uighur, Kirghiz, and Dai, revealed a distinct *NUDT15* genetic polymorphism pattern compared to the Han population. For example, the frequency of *NUDT15* common variations, rs746071566, was found to be lower in the Uighur and Kirghiz population than in the Han population. Additionally, rare disruptive variants (c.137C > G and c.138T > G) were identified in a Uighur child, highlighting the genetic diversity that could influence thiopurine toxicity. While data on the African population is still limited, existing research showed that *NUDT15* variants are rarely found in this population, making *TPMT* a more significant toxicity predictor. These findings underscore the need to expand pharmacogenetics research beyond the main ethnic groups to provide truly personalized medicine [86].

In summary, the existing body of evidence, despite showing a promising signal, especially in the Asian population, reveals limitations and discrepancies in our knowledge of *NUDT15* pharmacogenetics across diverse ethnicities and disease conditions. The strong association between specific *NUDT15* variants and severe myelotoxicity warrants consideration for pre-emptive genotyping, notably in populations with higher risk allele frequencies. This is particularly evident in Asian populations, where the *NUDT15* variants explain a significantly larger proportion of thiopurine-related toxicity and dose-reduction needs compared to the *TPMT* variants, which are the main genetic determinant in European cohorts. The success of initial RCTs on patients with IBD and recent positive findings in pediatric patients with ALL indicate that genotype-guided dosing is a viable strategy to enhance the safety of thiopurine drugs. On the other hand, large-scale prospective studies, involving various ethnic groups and focusing on disease-specific contexts, such as pediatric ALL, are pivotal to refining our knowledge and to developing universally applicable clinical guidelines. Limited data beyond the Asian cohorts with IBD highlights a significant area for future study to comprehensively embody the potential of *NUDT15* pharmacogenetics in personalized thiopurine therapy globally.

## 5. Challenges and Future Perspectives 

Having reviewed in vitro and in vivo foundational studies, along with observational studies and randomized controlled trials, which have highlighted the clinical impact of *NUDT15* variants, it becomes evident that, despite the significant advancements achieved, substantial gaps in research and knowledge persist. Addressing these challenges is crucial to comprehensively harness the potential of *NUDT15* pharmacogenetics in personalized thiopurine therapy. The future trajectory of this field will chiefly focus on preclinical and clinical research.

### 5.1. Preclinical Studies: Expanding Mechanistic Understanding 

Preclinical investigations remain pivotal in unraveling the whole functional picture of *NUDT15* and the precise mechanism on how its variants induce thiopurine intolerance. One of the previously highlighted main challenges is the incomplete characterization of the full spectrum of *NUDT15* apart from the common variants, such as p.Val18Ile and p.Arg139His, or other haplotypes. Future preclinical studies should focus on comprehensive functional examinations, e.g., high-throughput screening, as demonstrated by Suiter et al. [67], to accurately determine the enzymatic activity and/or structural stability of the less-studied or rare variants. These studies will provide a more comprehensive genotype–phenotype map, which is essential for translating preclinical findings to diverse patient populations.

Beyond characterizing *NUDT15* variants, another interesting area to be explored is the development of *NUDT15* inhibitors. Though the therapeutic application of the inhibitor to modulate the metabolism of thiopurines in malignancies is a compelling concept, current preclinical literature mainly highlights its use in the context of research, such as in structural studies aimed at providing a clearer understanding of the enzyme’s active site and catalytic mechanism [87]. The potential of these substances to enhance the efficacy of thiopurine therapy by inhibiting the detoxifying function of NUDT15 represents a promising, albeit early-stage, avenue for future preclinical studies.

Furthermore, an unanswered critical question from a mechanistic standpoint is the precise extent to which the 6-MP dose needs to be decreased in individuals harboring the *NUDT15* variants and the foundational reasoning behind this precise decrease. While clinical trials have shown that a lower initial dose is effective in suppressing toxicity in patients with *NUDT15* variants, the rigorous biochemical or cellular pathways dictating this dose–response curve in variant carriers have not been fully understood. Future preclinical studies, potentially using advanced cellular models or refined *Nudt15* knock-in mice models, could explore the complex metabolic flux and cellular response to various thiopurine doses in the presence of specific *NUDT15* defects. Further mechanistic insights will provide the foundational knowledge required for developing highly precise and evidence-based dose-adjustment algorithms in clinical settings. Potential compensatory pathways, as previously discussed regarding the broader NUDT family, also require further preclinical investigation to understand whether and how other enzymes affect *NUDT15* deficiency, which could influence overall toxicity.

### 5.2. Clinical Studies: Refining Precision and Expanding Generalizability 

On the clinical front, several critical challenges require collaborative effort to bridge the existing knowledge gap and to ensure an equitable application of *NUDT15* pharmacogenetics. Firstly, a pervasive limitation in various observational studies is the domination of studies in adult patients with IBD, particularly in the Asian population. Despite offering valuable evidence, the generalizability of their findings to other ethnic groups and, most importantly, to pediatric patients with ALL, remains a significant challenge. A contrasting study, such as that by Correa-Jimenez et al. [78] in Colombian patients with ALL, despite the small sample size, highlights the ethnic variability and the need for dedicated research in non-Asian cohorts. Large-scale observational prospective studies and RCTs in the future are urgently needed in various ethnic groups, including Asian, Caucasian, and Latin American populations, to ensure the prevalence and clinical impact of *NUDT15* variants specific to these ethnic groups.

Crucially, there is a pressing need for more dedicated studies in pediatric patients with ALL. While the recent multicenter RCT study by Zhou et al. [82] offers a solid foundation in this specific population, the overall volume of evidence is much smaller compared to IBD. Future clinical studies in pediatric patients with ALL should aim to establish a robust and evidence-based genotype-guided dosing guideline, mitigating not only myelosuppression but also other potential adverse drug reactions (ADRs).

Another significant area for advancement lies in the integration of the monitoring of metabolite levels with the *NUDT15* genotype, especially through RCTs. While several observational studies, such as Kang et al. [79] and Ju et al. [33], have employed the metabolite level as a surrogate marker, there remains a gap in RCTs that simultaneously measure metabolite levels (for instance, 6-TGN) and clinical outcomes such as myelosuppression or other ADRs. Future RCT studies should incorporate this dual measurement to definitively determine the association between the surrogate marker and the final clinical outcome. Such an established association will help in defining whether genotype-guided dosing actually optimizes the metabolite levels and whether the optimized level reliably reduces toxicity without losing efficacy. In addition, our knowledge regarding the association between metabolite levels and *NUDT15* variants will help mitigate the high variability observed in the DNA-TGN levels among variant carriers, as seen in Ju et al. [33], suggesting that a more dynamic and integrated approach may be needed.

To advance the application of *NUDT15* pharmacogenetics globally, our efforts should be focused on the development of standardized clinical guidelines that count for ethnic variability and disease-specific context. These efforts require international collaboration to harmonize testing methodologies, determine clinically actionable thresholds for different genotypes, and provide clear recommendations on dose adjustment. Furthermore, the cost-effectiveness of *NUDT15* genotyping poses a practical challenge for widespread clinical application, particularly in resource-limited settings, due to barriers such as limited access to advanced technology and insufficient funding. Innovative solutions such as pooled regional testing are therefore essential for mitigating these obstacles and enhancing accessibility.

Another critical area for future study is to conduct more extensive research in less-represented yet diverse populations, such as pediatric patients with ALL in low- and middle-income countries and specific ethnic groups such as Africans and Uighur/Kirghiz minorities to address the current knowledge gap. This is crucial for providing a truly comprehensive pharmacogenetic understanding which urgently needs strong supporting data to inform the development of personalized thiopurine therapy at a more local level. Future research should prioritize countries with a very genetically diverse population and significant pediatric ALL burden, which urgently needs strong supporting data to mitigate the existing knowledge gap. A commitment to inclusive studies will be crucial in ensuring that the benefits of *NUDT15* pharmacogenetics reach all patients in need and fully realize the promise of personalized medicine.

Finally, the findings from various studies in this review underscore that the most optimal thiopurine therapy strategy necessitates an integrated approach. This can be manifested through a step-wise clinical algorithm that combines the strength of pre-emptive pharmacogenetics screening (by focusing on *NUDT15* and *TPMT* variants), dynamic monitoring of metabolite level (e.g., 6-TGN level), and continuous dose adjustment based on the patient’s clinical response. This multimodal approach allows clinicians to utilize the power of predictive genotyping with real-time data from therapeutic drug monitoring, thus creating the most personalized, effective, and safe therapy for patients.

## Figures and Tables

**Figure 1 medsci-13-00112-f001:**
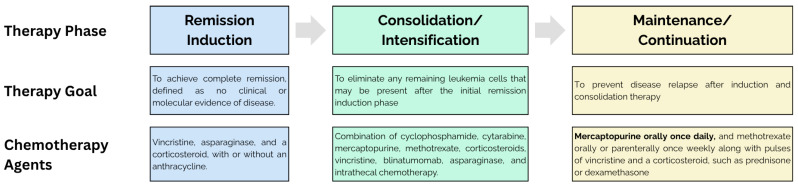
Overview of pharmacological therapy for acute lymphoblastic leukemia (ALL), detailing the three distinct phases (induction, consolidation, and maintenance), with their respective goals and commonly used chemotherapy agents.

**Figure 2 medsci-13-00112-f002:**
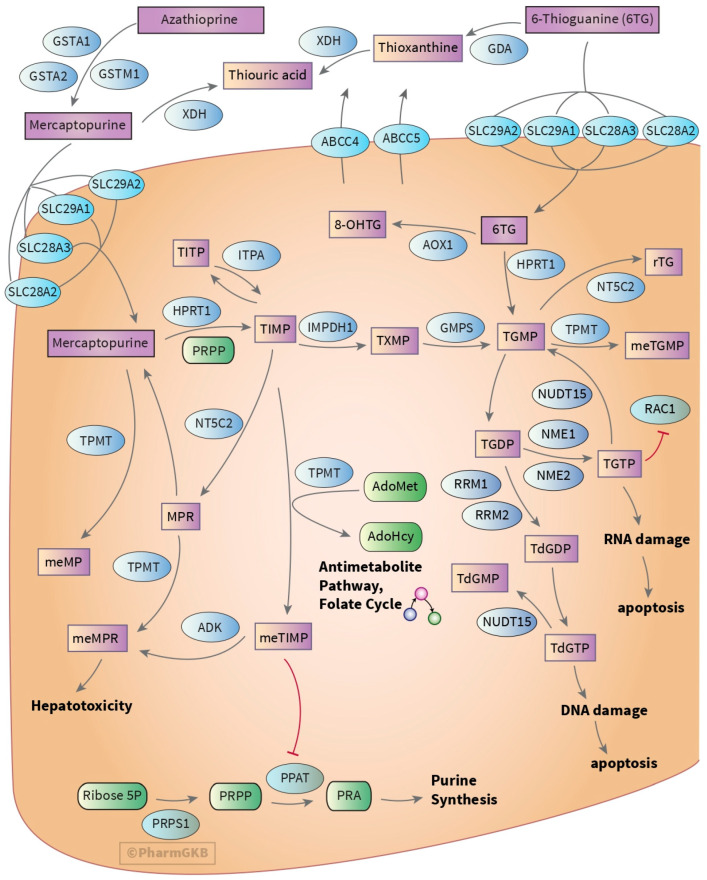
Diagrammatic representation of a non-tissue-specific cancer cell displaying candidate genes involved in the metabolism and action of thiopurines, including *NUDT15* [49,50].

**Figure 3 medsci-13-00112-f003:**
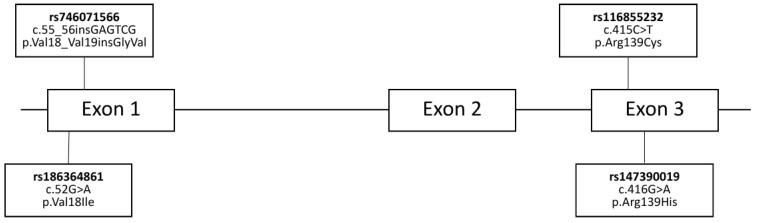
Schematic representation of the *NUDT15* gene structure and common pharmacogenetic variants. This diagram illustrates the three exons of the *NUDT15* gene and the positions of key single nucleotide polymorphisms (SNPs), including their rsIDs, corresponding DNA (c.) changes, and predicted amino acid (p.) alterations.

## Data Availability

The data that support the findings of this study are available from the corresponding author, Z.Z., upon reasonable request.

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
