# Peer review of "NUDT15 Pharmacogenetics in Acute Lymphoblastic Leukemia: Synthesizing Progress for Personalized Thiopurine Therapy"

_medsci, 2025, doi:10.3390/medsci13030112_

Round 1

Reviewer 1 Report

Comments and Suggestions for Authors

This review is focused on the biological mechanism and clinical significance of nudix hydrolase 15 (NUDT15) pharmacogenetics in thiopurine therapy of acute lymphoblastic leukemia (ALL). The authors provide a thorough, updated review of the role of NUDT15 in the detoxification of active thiopurines, along with the common genetic variants, which significantly disrupt activity of NUDT15, leading to the accumulation of toxic metabolites. The authors emphasize recent clinical data, which determined that NUDT15 genotype-guided dosing effectively reduce thiopurine-induced toxicity without interfering with therapeutic outcome.  The review highlights challenges in the field, which include the incomplete characterization of rare NUDT15 variants, and the need for standardized integration with metabolite monitoring. The authors conclude that NUDT15 pharmacogenetics is essential for improving patient safety and thiopurine dosage optimization in the treatment of ALL, and that future research should focus on increasing genetic data across different populations, improving the dose adjustment algorithm, and harmonizing therapeutic guidelines.

This is an outstanding manuscript that focuses on the mechanisms that regulate efficacy and toxicity of thiopurine therapy in ALL. The manuscript is well-written, and easily understandable. The review covers an important topic that is of high interest for readers of Medical Sciences. The role of NUDT15 in regulation of thiopurine therapy has been largely overlooked, which increases the significance and the overall impact of this manuscript. Outstanding figure that displays regulatory pathways is a strength of the manuscript. This figure ensures that this manuscript will be often used as a reference by the readers. Overall, this is an excellent, high-impact paper. There are a few minor concerns, which should be addressed in order to provide more comprehensive overview of NUDT15 in therapy of ALL. In short: 1) while the authors appropriately focus on the role of NUDT15 polymorphism in regulation of thiopurine therapy, they should also briefly mention that the inhibitors of NUDT15 were tested preclinically to target thiopurine pathway in human malignancies; 2) The authors should briefly mention that n addition to polymorphism, alterations in NUDT15 transcription and expression have been associated with resistance to 6-mercaptopurine treatment. The authors should also mention a potential role of microRNA in regulation of NUDT15 expression; and 3) While in introduction, the authors correctly mentioned that the initial focus of the thiopurine therapy was on the polymorphism of thiopurine methyltransferase (TPMT), they should also mention that the extensive research was done on the role and polymorphism of cytosolic 5'-nucleotidase II (NT5C2) gene in thiopurine therapy.

In summary, this is an outstanding manuscript that focuses on the role of NUDT15 pharmacogenetics in thiopurine therapy of ALL.  The above-outlined strengths of the paper make this manuscript particularly suitable for publication in Medical Sciences. Addressing the minor concerns outlined above, will improve the clarity and overall impact of the manuscript.  

Reviewer 2 Report

Comments and Suggestions for Authors

I sincerely appreciate the opportunity to evaluate this comprehensive review, which addresses the critical role of NUDT15 pharmacogenetics in mitigating thiopurine toxicity during acute lymphoblastic leukemia (ALL) therapy. The manuscript effectively synthesizes emerging evidence on genotype-guided dosing, particularly in Asian populations, and highlights the clinical validation of NUDT15 testing through recent RCTs. However, to elevate its impact and translational relevance, I recommend major revisions to address key gaps, as outlined below.

  1. Add a dedicated section detailing NUDT15’s structural biology, enzymatic function (e.g., hydrolysis of TGTP/TdGTP), and functional impact of key variants (e.g., p.Arg139Cys, p.Val18_Val19insGlyVal). Include in vitro/knockout model data to underscore metabolite accumulation and DNA damage mechanisms.
  2. Clarify NUDT15’s relative contribution versus TPMTusing quantitative metrics (e.g., odds ratios: NUDT15 variants confer 28% dose reduction needs vs. 10% for TPMT in Asians). Discuss ethnic disparities NUDT15 explains 22% of toxicity variance in Asians vs. TPMT in Europeans.
  1. Integrate meta-analysis data from RCTs (e.g., 50% dose reduction for NUDT15 heterozygotes; homozygous variants tolerate only 8% standard doses). Highlight CPIC’s 2024 update on dual *TPMT/NUDT15* intermediate metabolizers (20–50% dose reduction).
  2. Explore ITPA’s role in thiopurine intolerance, especially in Middle Eastern cohorts where ITPAvariants correlate with 65% dose reductions. Address *NUDT15/TPMT* compound heterozygosity (0.9% in Brazilians 13), requiring aggressive dose adjustments.
  3. Expand on understudied populations (e.g., Africans, Uighur/Kirghiz minorities with rare NUDT15variants). Propose a stepwise clinical algorithm combining NUDT15 genotyping, metabolite monitoring (e.g., TGN levels), and dynamic dosing.
  4. Discuss cost-effectiveness of preemptive testing, citing CPIC’s tiered dosing recommendations. Address barriers in low-resource settings (e.g., limited genotyping access) and solutions like pooled regional testing.

Reviewer 3 Report

Comments and Suggestions for Authors

I have had a long time interest in thiopurine metabolism applied to improving ALL therapy.  Your manuscript was a pleasure to read.    It is  a concise and clear presentation of an extremely comprehensive review of the subject.  

I have no concerns and no corrections.   Great job!

Author Response

We sincerely thank the reviewer for their time and for their very kind and encouraging feedback. We are delighted that the manuscript was a pleasure to read and that our efforts to provide a concise, clear, and comprehensive review of the subject were well-received. Your long-standing interest in thiopurine metabolism makes your positive comments particularly valuable to us.

Round 2

Reviewer 2 Report

Comments and Suggestions for Authors

The authors addressed my comments very carefully